# GRAD-TOPOCAM: EEG BRAIN REGION VISUAL INTERPRETABILITY VIA GRADIENT-BASED TOPOGRAPHIC CLASS ACTIVATION MAP

## ABSTRACT

The visualization and interpretability of electroencephalogram (EEG) decoding significantly contribute to brain-computer interfaces (BCI) and cognitive neuroscience. Although some existing research has attempted to map EEG features to specific brain regions, these approaches fail to fully utilize raw signals and lack extensibility to other Deep Learning (DL) models. In this work, Grad-TopoCAM (Gradient-Based Topographic Class Activation Map) is proposed, which enhances interpretability in DL models for EEG decoding adaptively. Grad-TopoCAM calculates the gradient of feature maps for the target class at the target layer. The weights of the feature maps are obtained through global average pooling of the gradients. The class activation map is generated by performing a linear combination of weights and feature maps, which is subsequently mapped to different brain regions. Grad-TopoCAM is validated across eight DL models on four public datasets. Experimental results indicate that Grad-TopoCAM effectively identifies and visualizes brain regions that significantly influence decoding outcomes, while also facilitating channel selection for different decoding tasks. The code and data are open-source.

## 1 INTRODUCTION

Electroencephalogram (EEG) decoding is the cornerstone of brain-computer interfaces (BCI) (Ji et al., 2024). The introduction of interpretability and visualization methods not only enhances the transparency and reliability of models but also promotes deeper exploration in neuroscience and clinical applications (Miao et al., 2023). By employing such methods, researchers can gain insight into which brain regions the model emphasizes during decision-making processes, thereby highlighting the critical roles of specific brain regions in brain activity and offering valuable guidance for future neuroscience research (Zong et al., 2024).

Despite the remarkable progress of Deep Learning (DL) as an end-to-end "black-box" method in various fields, (Phan-Trong et al., 2023) its inherent opacity poses significant challenges to interpretability. While interpretability techniques such as Grad-CAM (Selvaraju et al., 2017) and LIME (Ribeiro et al., 2016) have been extensively applied in Computer Vision (CV) and Natural Language Processing (NLP), their use in EEG signal decoding remains underexplored. Current EEG decoding research predominantly relies on complex, indirect approaches for interpretability analysis (Sujatha Ravindran & Contreras-Vidal, 2023). In some studies, EEG signals are transformed into two-dimensional feature maps (Qian et al., 2024), (Ding et al., 2023), or multi-channel signals are mapped into two-dimensional matrices (Li et al., 2020), followed by visualization with Grad-CAM. However, these methods struggle to reveal the specific brain regions that deep models focus on during decision-making. Though some studies have designed specific algorithms for proposed methods to visualize brain region features (Cai & Zeng, 2024), these approaches generally lack generalizability and are not easily adaptable for feature visualization across any target network layer. Consequently, current interpretability and visualization methods of EEG decoding still lack a universal interpretability method that can directly map model-decision features to corresponding brain region activity.

In this work, we propose a universal interpretability and visualization method, Grad-TopoCAM (Gradient-Based Topographic Class Activation Map), which directly maps EEG features to brain regions in EEG decoding. When the raw EEG signals are input into the DL model, Grad-TopoCAM computes the gradients of the feature maps at the target layer for the predicted class. The gradients are then globally averaged to calculate the feature map weights. By linearly combining the weights with the feature maps, class activation maps are generated for the target class. The class activation maps are subsequently mapped to different brain regions, illustrating the varying contributions of each brain area in EEG decoding.

We summarize our contributions below.

1. We propose Grad-TopoCAM, a class-discriminative localization technique that generates visualizations of salient brain region features from DL models without requiring modifications to the architecture or retraining.

2. Grad-TopoCAM has been validated across eight different DL models and four publicly available datasets, with the salient brain features aligning with established findings in cognitive neuroscience.

3. Grad-TopoCAM is applied to the multi-layer convolutional structure of the EEGNet network. As the convolutional layers of EEGNet deepen, Grad-TopoCAM reveals the feature variations of different brain regions in the EEG decoding decision-making process.

4. The visualizations of salient brain region features generated by Grad-TopoCAM can be utilized to identify key brain areas, facilitating EEG channel selection.

The remainder of this article is organized as follows. Section 2 describes the related work. Section 3 describes the proposed method. Section 4 presents the datasets, DL models, and brain topography. Section 5 describes the discussion. Finally, Section 6 presents the conclusion and future work directions.

## 2 RELATED WORKS

Early EEG visualization methods primarily rely on topographic maps (Vafaei et al., 2023), (Cline et al., 2023) and time-frequency (Cai et al., 2022), (Kiselev et al., 2022) representations. Although these methods display the characteristics of raw EEG signals (Ding et al., 2023), (Currey et al., 2023), (Shi et al., 2024), they fail to reveal the brain regions that play a critical role in EEG decoding. With the introduction of DL, researchers explore how to apply the interpretability of DL models to EEG signal decoding. In the existing studies, Li et al. (2022) convert EEG signals into brain topography images, train these images, and employ Grad-CAM for visualization. Nevertheless, this method does not fully utilize the raw signals, resulting in limitations in interpretability. Moreover, Qian et al. (2024) utilize Grad-CAM to visualize features on the time-frequency representation of EEG signals but are unable to accurately identify the brain regions that significantly contribute to the results. This shortcoming renders the interpretation of the model insufficient. To enhance the direct utilization of raw EEG signals, Li et al. (2020) and Cui et al. (2022) propose mapping multi-channel signals into two-dimensional matrices and inputting them into a two-dimensional convolutional neural network (CNN) for training. This approach aims to achieve image-like feature visualization of the matrices, ultimately generating visualization results. Even so, while this method somewhat improves interpretability, it remains limited by the necessity of employing a two-dimensional convolutional structure within the neural network. To overcome this limitation, Cai & Zeng (2024), Song et al. (2022) and Miao et al. (2023) propose various EEG decoding models that simultaneously enable feature visualization across specific network layers corresponding to different brain regions. Despite these advancements, these feature visualization methods require additional specific design, limiting their applicability for direct feature mapping across different target network layers and lacking generalizability and flexibility. Therefore, exploring universal interpretability and visualization methods that effectively link model features to brain region activity is essential.

## 3 METHOD

In this work, we propose a novel and generalizable interpretable visualization method, named Grad-TopoCAM, as shown in Figure 1. The proposed method aims to directly map salient features from

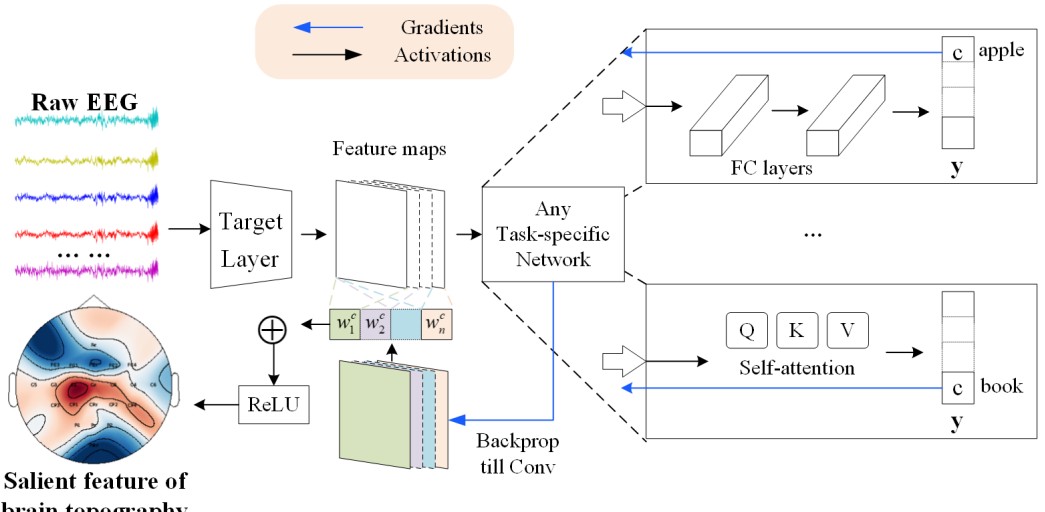

Figure 1: Architecture of the proposed Grad-TopoCAM.

target layers onto brain topographs, thereby identifying key brain regions involved in EEG decoding. Unlike existing indirect and complex methods, Grad-TopoCAM facilitates feature-brain region mapping at any target layer, significantly enhancing the universality and flexibility.

Grad-TopoCAM calculates the class activation map for the target layer of a specific class. The detailed steps are as follows:

1. Forward Propagation: EEG signals are input into the model to obtain the predicted probability scores $y^c$ for each class, where $c$ represents the class index.

2. Calculation of Gradient: To reveal the brain regions that the network focuses on during the classification process, we perform backpropagation on the gradient of the predicted score $y^c$ for the target class with respect to the feature maps $A^k$ of the target layer. The gradient for each feature map in the network is calculated, represented as $\frac{\partial y^c}{\partial A^k}$, where $k$ denotes the index of the $k^{th}$ feature map. Additionally, the target layer can be other layer within the network architecture, such as convolutional layers, self-attention layers, or batch normalization layers.

3. Calculation of Feature Map Weights: The gradients are processed through global average pooling to derive the weights for each feature map, represented as $\alpha_k^c$. These weights reflect the significance of the feature map $A^k$ in contributing to the prediction of class $c$:

$$\alpha_k^c = \frac{1}{Z} \sum_i \sum_j \frac{\partial y^c}{\partial A_{ij}^k} \tag{1}$$

where $Z$ represents the size of the feature map, and $i$ and $j$ index the corresponding spatial locations within the feature map.

4. Calculation of salient Feature Values: The weights $\alpha_k^c$ and the feature maps $A^k$ are linearly combined to generate the heatmap $L^c$ for class $c$, as expressed by the following equation:

$$L^c = \text{ReLU}\left( \sum_k \alpha_k^c A^k \right) \tag{2}$$

where ReLU function is applied to ensure that negative values within the heatmap are set to zero, preserving only the positive values. This process retains only the regions that contribute positively to the target classification, thereby facilitating the precise localization of the most influential areas within the brain during the decoding process.

5. Mapping of Salient Feature Values to Brain Topography: The salient feature values for each EEG channel are averaged to generate the brain topographic. The subsequent formula is as follows:

$$L_{avg}^c = \frac{1}{T} \sum_{t=1}^{T} \left( \text{ReLU} \left( \sum_k \alpha_k^c A^k \right) \right) = \frac{1}{T} \sum_{t=1}^{T} L_t^c \tag{3}$$

where $T$ denotes the dimensionality of the salient feature values.

## 4 EXPERIMENT

### 4.1 DATASETS AND PREPROCESS

Dataset I: The BCI Competition IV Dataset 2a (Tangermann et al., 2012), provided by Graz University of Technology, contains EEG recordings from nine healthy participants. The EEG signals were acquired using a 10–20 electrode system with 22 Ag/AgCl electrodes, sampled at 250 Hz. Each participant was instructed to perform four distinct motor imagery tasks: imagining movements of the left hand, right hand, both feet, and tongue. The data was filtered to [4, 40] Hz using a band-pass filter.

Dataset II: Nieto et al. (2022) developed an inner speech EEG dataset consisting of 10 native Spanish-speaking participants. EEG recordings were acquired using the 10-20 system with 128 EEG channels and 8 external EOG/EMG channels at a sampling rate of 1024 Hz. Participants were instructed to silently articulate four Spanish words: "arriba" (up), "abajo" (down), "derecha" (right), and "izquierda" (left). During preprocessing, the data were re-referenced using earlobe channels and filtered with a band-pass filter ranging from 0.5 to 100 Hz, along with a notch filter at 50 Hz. The sampling rate was then downsampled to 254 Hz. Independent component analysis (ICA) was employed to remove artifacts, ensuring signal quality.

Dataset III and Dataset IV: Li et al. (2024) developed a silent reading EEG dataset, which includes data from a single participant who is a native Mandarin speaker and proficient in English as a second language. The EEG activity was recorded over 26 days while the participant silently read seven Chinese words and nine English words. The Chinese words included: "你", "去", "天", "头", "来", "水", and "说", while the English words were: "apple", "book", "come", "cup", "go", "head", "stand", "water", and "you". EEG signals were collected using 64 electrodes (with 59 EEG channels and 5 body function channels) based on the 10-20 system, at a sampling rate of 1000 Hz. Dataset III consists of Chinese words and Dataset IV consists of English words

### 4.2 OVERVIEW OF DL MODELS

Eight different DL models are employed to train and test across four distinct datasets. The models are summarized below.

ConvNet (Schirrmeister et al., 2017) family includes multiple convolution and pooling layers. ShallowConvNet uses a single convolutional layer, while DeepConvNet leverages multiple convolutional layers to capture more complex features.

EEGNet (Lawhern et al., 2018) is a compact CNN model. It consists of three key components: one-dimensional convolution for temporal feature extraction, depthwise separable convolution for spatial feature learning, and a fully connected layer for classification.

RACNN (Fang et al., 2020) is a novel regional attention convolutional neural network that extracts spectral-spatial-temporal features. It aggregates spectral-temporal features produced by a convolutional neural network into fixed-length features.

EEG-ChannelNet (Palazzo et al., 2020) consists of a series of convolutional modules that initially extract temporal and spatial features using 1D convolutions. These features are refined through residual layers, with final predictions generated via convolutional and fully connected layers.

Conformer (Song et al., 2022) is a compact Convolutional Transformer model. It captures both local and global features using convolutional layers and self-attention modules. A simple fully connected classifier is then employed to predict EEG categories.

Table 1: Classification accuracy across different models in Dataset I.

| Model | S01 | S02 | S03 | S04 | S05 | S06 | S07 | S08 | S09 |
|---|---|---|---|---|---|---|---|---|---|
| ShallowConvNet | 73.91% | 42.36% | 85.42% | 61.11% | **47.92%** | 46.53% | 85.42% | 75.69% | 73.26% |
| DeepConvNet | 43.4% | 39.24% | 47.92% | 46.18% | 34.38% | 39.24% | 46.18% | 44.79% | 55.55% |
| EEGNet | 78.81% | **53.12%** | 82.64% | 58.33% | 40.62% | 44.1% | 68.06% | 74.31% | 72.22% |
| RACNN | 40.97% | 31.25% | 38.54% | 33.33% | 32.99% | 34.72% | 35.42% | 38.19% | 47.92% |
| EEG-ChannelNet | 40.62% | 27.43% | 48.96% | 43.75% | 28.82% | 35.42% | 38.19% | 40.62% | 52.43% |
| Conformer | **81.6%** | 51.73% | **90.62%** | **71.53%** | 34.38% | **52.78%** | **88.53%** | 79.51% | **79.17%** |
| LMDA-Net | 57.29% | 32.99% | 68.06% | 47.22% | 34.03% | 40.62% | 39.93% | 46.53% | 64.92% |
| D-FaST | 64.92% | 36.8% | 70.49% | 47.57% | 34.72% | 42.36% | 64.24% | 65.97% | 68.06% |

Table 2: Classification accuracy across different models in Dataset II.

| Model | S01 | S02 | S03 | S04 | S05 | S06 | S07 | S08 | S09 |
|---|---|---|---|---|---|---|---|---|---|
| ShallowConvNet | 40.0% | **35.0%** | **42.5%** | 35.0% | 40.0% | 20.0% | 40.0% | **40.0%** | 35.0% |
| DeepConvNet | 37.5% | 27.5% | 37.5% | 32.5% | **45.0%** | 40.0% | 35.0% | 37.5% | 40.0% |
| EEGNet | **42.5%** | 27.5% | 40.0% | 35.0% | 37.5% | 32.5% | 35.0% | 32.5% | 35.0% |
| RACNN | **42.5%** | **35.0%** | 40.0% | **37.5%** | 37.5% | **45.0%** | **42.5%** | 35.0% | 37.5% |
| EEG-ChannelNet | 25.0% | 22.5% | 35.0% | 20.0% | 30.0% | 35.0% | 25.0% | 35.0% | 32.5% |
| LMDA-Net | 32.5% | **35.0%** | 27.% | **37.5%** | 37.5% | 35.0% | 30.0% | 32.5% | **45.0%** |
| D-FaST | 37.5% | 30.0% | 32.5% | 30.0% | 37.5% | 40% | 40% | 32.5% | 32.5% |

Table 3: Classification accuracy across different models in Dataset III and Dataset IV.

| Model | Chinese | English |
|---|---|---|
| ShallowConvNet | 12.36% | 9.09% |
| DeepConvNet | **17.98%** | **19.00%** |
| EEGNet | 10.11% | 14.05% |
| EEG-ChannelNet | 15.73% | 17.36% |
| LMDA-Net | 14.61% | 10.74% |
| D-FaST | 12.36% | 11.57% |

LMDA-Net Miao et al. (2023) is a lightweight multi-dimensional attention network that integrates channel and depth attention modules to efficiently extract features across multiple dimensions.

D-FaST (Chen et al., 2024) is a novel Disentangled Frequency-Spatial-Temporal Attention model. It consists of three key components: multi-view attention for frequency domain features, spatial extraction via dynamic brain connection graph attention, and temporal features through a local sliding window attention mechanism.

### 4.3 CLASSIFICATION ACCURACY AND BRAIN TOPOGRAPHY

The performances on different datasets across multiple models are shown as Tabel 1, Table 2, Table 3. In Dataset I, the Conformer model achieved the highest accuracy, particularly excelling for subjects S03, S06, and S09, with rates of 90.62%, 88.53%, and 79.17%. EEGNet performed well for S03 and S09 but was generally outperformed by Conformer. ShallowConvNet showed similar results, performing best on S03, S07, and S09 with accuracies of 85.42%, 85.42%, and 73.26%. In Dataset II, RACNN was the most stable, outperforming EEGNet for S01, S06, and S09. Shallow-ConvNet had moderate success for some subjects, though overall accuracy was lower. For Dataset III and Dataset IV, accuracy was generally low, with DeepConvNet performing slightly better than the others, achieving 17.98% and 19.00%, respectively.

Based on the classification accuracy results, the proposed Grad-TopoCAM is employed for visualization analysis on the model with the highest accuracy for each subject. The contributions of different brain regions to the model's decisions are displayed.

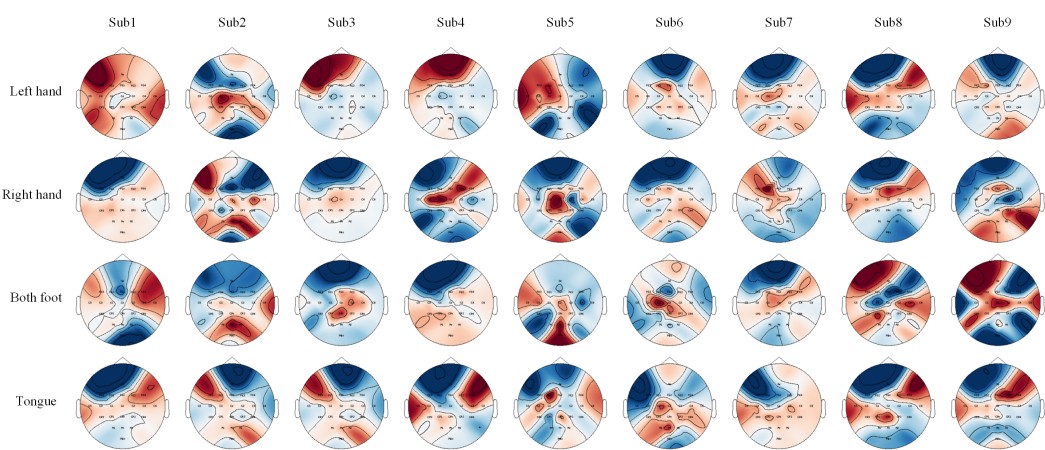

Figure 2: Salient feature of brain topography in Dataset I.

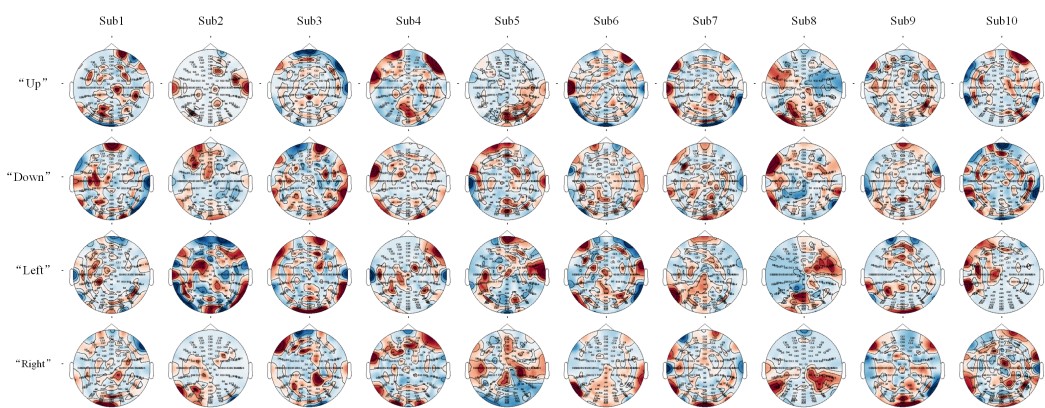

Figure 3: Salient feature of brain topography in Dataset II.

In Dataset I (motor imagery), as shown in Figure 2, central regions (C3, Cz, CPz) demonstrated significant contributions across multiple subjects. This aligns with existing research that identifies motor-related areas, especially C3 and Cz, as key regions in motor imagery tasks (Bai et al., 2007),(Yon et al., 2018), (Wang et al., 2023). Additionally, channels like C5, CP1, and FC2 show high contributions in some subjects, suggesting that both precentral and parietal regions play crucial roles in the decision-making of classification models during motor imagery tasks.

In Dataset II (inner speech), as shown in Figure 3, the frontal areas (A19, D32, B7) and parietal regions (A21, D17) contribute significantly. These regions are associated with complex cognitive processes and perceptual integration, with the prefrontal cortex playing a key role in speech generation and understanding, and the parietal lobe being involved in spatial and linguistic integration (Friederici, 2011), (Fedorenko et al., 2024). The high contributions from these regions suggest that inner speech relies on higher-order cognitive functions like attention, working memory, and visual processing, consistent with previous cognitive task findings in EEG research.

In Datasets III (Chinese words) and Datasets IV (English words), as shown in Figure 4 and Figure 5, visual-related regions (Oz, POz, PO8) and frontal areas (AF3, Fp1, Fp2) contribute significantly in word classification tasks across multiple subjects. This indicates that visual and frontal regions play a central role in language comprehension and processing, highlighting the close connection between visual representation and linguistic cognition during word processing tasks (Kutas & Federmeier, 2000), (de Varda et al., 2024). Despite the linguistic differences between Chinese and English, the similar patterns of brain activation suggest common cognitive processing mechanisms (Liu et al., 2023), underscoring the deep neural underpinnings of language.

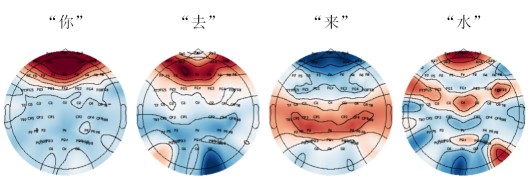

Figure 4: Salient feature of brain topography in Dataset III.

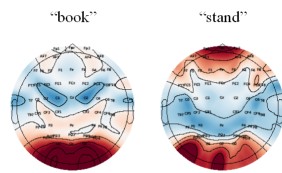

Figure 5: Salient feature of brain topography in Dataset IV.

## 5 DISCUSSION

### 5.1 LAYER-WISE BRAIN REGION FOCUS IN EEGNET

We observe the dynamic changes in brain regions across different network layers by the proposed Grad-TopoCAM, as shown in Figure 6. A layer-wise analysis of Datasets I (motor imagery), based on EEGNet, reveals how crucial brain areas become progressively focused as the convolutional layers deepen. For instance, in Layer0 for Label0, task-relevant regions exhibit a broader distribution, including areas such as CP2, CPz, and C4. However, by Layer2 and Layer3, the most contributive regions converge around Cz, CPz, and C1, demonstrating that deeper layers capture more task-specific features. This pattern is consistent across other true labels, where shallow convolutional layers show dispersed activations, and deeper layers focus on regions closely associated with motor control. This layer-wise feature visualization illustrates how EEGNet hones in on more precise task-relevant regions as the network deepens, validating the efficacy of the propposed Grad-TopoCAM.

### 5.2 CHANNEL SELECTION ON EFFICIENCY AND FERFORMANCE

Through Grad-TopoCAM visualization, the contribution of different brain regions to classification tasks can be determined. Channel rankings for each label are calculated based on their individual significance, and these rankings are weighted and summed to derive the final channel sequence for each subject. As shown in Table 4, "Full Channel Signals" refers to the use of all original signal channels, while "Selected Channel Signals" represents the use of the top half of the channels with the highest contributions.

Channel selection significantly optimizes both the parameter and computational demands of the models. For example, in EEGNet, the parameter complexity decreases from 130.245M to 59.175M, and the computational count is reduced from 213.748K to 86.772K. This reduction substantially lowers the computational burden while maintaining good performance, making real-time processing and application more feasible. Similarly, other models such as LMDA-Net and ShallowConvNet also show marked reductions in parameter and computation requirements, laying a foundation for practical deployment.

Channel selection not only improves computational efficiency but also enhances classification performance by focusing on brain regions that are most crucial for the task, as shown in Table 5. For instance, ShallowConvNet's accuracy for subject S06 increases by 20.0%, and DeepConvNet demonstrates consistent performance across multiple tasks. This suggests that the channel selected is beneficial for the classification performance. Additionally, D-FaST also achieves a balance between accuracy and computational efficiency in certain tasks. However, some models experience a drop in classification performance after channel selection. For instance, EEGNet's accuracy decreases from 64.175% with full channels to 59.175% with selected channels. This decline could be attributed to the complexity of EEG signals and the redundancy of brain region signals. While chan-

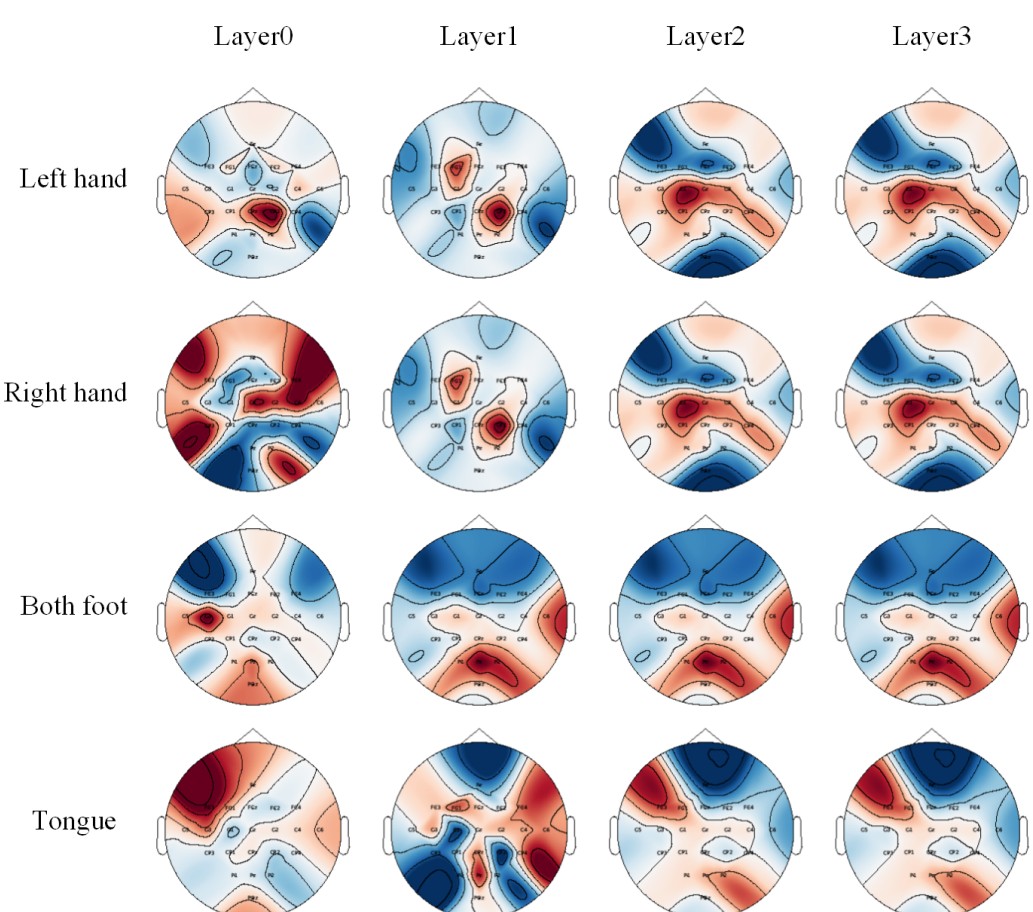

Figure 6: Salient features of brain topography across target layers in EEGNet models.

Table 4: Comparison of model Parameters and FLOPs before and after channel selection.

| Model | Full Channel Signals | | Selected Channel Signals | |
|---|---|---|---|---|
| | Params | FLOPs | Params | FLOPs |
| ShallowConvNet | 324.981M | 215.684K | 162.575M | 113.284K |
| DeepConvNet | 253.485M | 363.284K | 138.746M | 260.884K |
| EEGNet | 130.245M | 213.748K | 59.175M | 86.772K |
| EEG-ChannelNet | 23.202G | 20.090M | 11.596G | 6.582M |
| LMDA-Net | 288.759M | 8.388K | 144.396M | 7.940K |
| D-FaST | 13.168G | 12.153M | 6.620G | 6.296M |

nel selection aims to focus on the most relevant channels for the task, in some cases, the removed channels may still contain information beneficial to the model's decision-making process. Overall, channel selection through Grad-TopoCAM not only enhances model performance and efficiency but also improves interpretability.

## 6  CONCLUSION

In this work, we propose Grad-TopoCAM, an innovative method that enhances the interpretability of EEG decoding in DL models. By adaptively mapping the gradients of feature maps to specific brain regions, Grad-TopoCAM not only highlights the areas of the brain that significantly impact decoding outcomes but also facilitates informed channel selection across diverse EEG tasks. The comprehen-

Table 5: Classification accuracy after channel selection across different models (with change relative to full channel signals).

| Model | S01 | S02 | S03 | S04 | S05 | S06 | S07 | S08 | S09 | S10 |
|---|---|---|---|---|---|---|---|---|---|---|
| ShallowConvNet | 35.0% (-5.0%) | 37.5% (2.5%) | 30.0% (-12.5%) | 37.5% (2.5%) | 40.0% (0.0%) | 40.0% (20.0%) | 32.5% (-7.5%) | 30.0% (-10.0%) | 42.5% (7.5%) | 37.5% (2.5%) |
| DeepConvNet | 35.0% (-2.5%) | 35.0% (7.5%) | 32.5% (-5.0%) | 40.0% (7.5%) | 40.0% (-5.0%) | 32.5% (-15.0%) | 35.0% (5.0%) | 42.5% (5.0%) | 45.0% (5.0%) | 40.0% (2.5%) |
| EEGNet | 37.5% (-5.0%) | 25.0% (-2.5%) | 37.5% (-2.5%) | 32.5% (-2.5%) | 40.0% (2.5%) | 25.0% (-7.5%) | 35.0% (0.0%) | 37.5% (5.0%) | 40.0% (5.0%) | 32.5% (0.0%) |
| RACNN | 40.0% (-2.5%) | 42.5% (7.5%) | 35.0% (-5.0%) | 40.0% (2.5%) | 37.5% (0.0%) | 40.0% (-5.0%) | 40.0% (-2.5%) | 35.0% (0.0%) | 47.5% (10.0%) | 45.0% (2.5%) |
| EEG-ChannelNet | 37.5% (12.5%) | 25.0% (2.5%) | 42.5% (7.5%) | 22.5% (2.5%) | 40.0% (10.0%) | 35.0% (0.0%) | 20.0% (-5.0%) | 40.0% (0.0%) | 30.0% (10.0%) | 25.0% (-15.0%) |
| LMDA-Net | 32.5% (0.0%) | 32.5% (-2.5%) | 32.5% (5.5%) | 32.5% (-5.0%) | 32.5% (-5.0%) | 32.5% (-2.5%) | 32.5% (2.5%) | 32.5% (0.0%) | 50.0% (-12.5%) | 32.5% (15.0%) |
| D-FaST | 42.5% (5.0%) | 27.5% (-2.5%) | 30.0% (-2.5%) | 25.0% (-5.0%) | 35.0% (5.0%) | 25.0% (10.0%) | 35.0% (0.0%) | 42.5% (10.0%) | 32.5% (10.0%) | 32.5% (10.0%) |

sive validation of Grad-TopoCAM across eight DL models and four public datasets demonstrates its robustness and versatility, marking a significant advancement in the field of BCI and cognitive neuroscience.

Despite the significant contributions of Grad-TopoCAM, the limitation is also consideration. The current implementation primarily focuses on enhancing interpretability within supervised learning frameworks. As such, its effectiveness in unsupervised or semi-supervised contexts remains unexplored. Future research could investigate the adaptation of Grad-TopoCAM to these paradigms, potentially expanding its applicability to a broader range of EEG analysis tasks.

ACKNOWLEDGMENTS

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

## A  APPENDIX

You may include other additional sections here.

