# OpenReview forum: "Grad-TopoCAM: EEG Brain Region Visual Interpretability via Gradient-Based Topographic Class Activation Map"
_ICLR.cc/2025/Conference — Submitted to ICLR 2025_

### Official Review · Reviewer_Swr2 · 2024-10-27

**Soundness:** 2
**Presentation:** 1
**Contribution:** 1
**Rating:** 3
**Confidence:** 3

**Summary:**

To solve the issue that existing EEG interpretability researches fail to fully utilize raw signals and lack extensibility to other Deep Learning (DL) models, this paper proposes a novel framework, Grad-TopoCAM, to enhances interpretability in DL models for EEG decoding adaptively. Grad-TopoCAM has been validated across eight different DL models and four publicly available datasets, with the salient brain features aligning with established findings in cognitive neuroscience.

**Strengths:**

The motivation, enhancing interpretability in DL models for EEG decoding adaptively, is strong and interesting.

The proposed methods, Grad-TopoCAM, can generate visualizations of salient brain region features from DL models without requiring modifications to the architecture or retraining.

**Weaknesses:**

The reviewer has some concerns about the technical contributions of this paper. The proposed method is very simple. CAM is a highly classical method that has been thoroughly explored in other fields. This paper merely extends its application to the visualization of EEG brain region features, with limited technical innovation.

The experimental setup is not clearly delineated; for instance, the hyperparameters for training and testing each model are not thoroughly detailed, and the dataset partitioning method is not explicitly described.


The writing of this paper has significant room for improvement. Some unnecessary section titles, such as Acknowledgments and Appendices, should not be included. The tables are not aesthetically pleasing; why not omit the percentage sign (%)? The displayed brain region topology maps have low resolution, making the content difficult to discern. Why not use vector graphics to render the brain region topology maps?

**Questions:**

Please see the weakness.

---

### Official Review · Reviewer_oiZi · 2024-10-28

**Soundness:** 2
**Presentation:** 2
**Contribution:** 1
**Rating:** 3
**Confidence:** 5

**Summary:**

This paper introduces Grad-TopoCAM, a novel method designed for visualizing brain region activation in EEG decoding using gradient-based localization. The primary goal is to enhance the interpretability of deep learning models applied to EEG data by directly mapping feature maps generated by these models to specific brain regions. However, the effectiveness of this visualization method has not been thoroughly validated. The experiments presented largely focus on evaluating the performance of different EEG decoders, resulting in unclear contributions from this research.

**Strengths:**

Grad-TopoCAM represents a significant effort to improve the interpretability of deep learning models in the context of EEG data.

The method's approach to directly mapping feature maps to brain regions offers a novel perspective on understanding model outputs and could facilitate advancements in neurotechnology research.

**Weaknesses:**

The validation of the effectiveness of the visualization method is insufficiently addressed, limiting the overall impact of the research.

The experiments mainly assess the performance of various EEG decoders without establishing the unique contributions of Grad-TopoCAM.

A comparison with established post-hoc explanation techniques, such as Grad-CAM and SmoothGrad, is lacking, which would help contextualize Grad-TopoCAM's performance and effectiveness.

**Questions:**

What specific metrics will be used to evaluate the effectiveness of Grad-TopoCAM compared to existing visualization techniques?

How do the authors plan to quantify the significance of the feature attributes revealed by Grad-TopoCAM in relation to classification accuracy?

---

### Official Review · Reviewer_RxwE · 2024-10-31

**Soundness:** 1
**Presentation:** 3
**Contribution:** 2
**Rating:** 3
**Confidence:** 4

**Summary:**

The paper proposed Grad-TopoCAM, which is an explainable AI method to identify and visualize brain regions that significantly influence decoding outcomes. The method was evaluated on multiple EEG datasets and provided with visualizations.

**Strengths:**

1. The paper attempted to address the explainability issues for EEG deep learning research which is a key gap in the field.
2. The paper has good structure and clarity of writing in general.
3. The figures are informative and clear.

**Weaknesses:**

1. In the related work section, it is unclear why 'employing a two-dimensional convolutional structure' is a limitation as this is a common approach for most of the works in the EEG field.
2. The key weakness is that there is no comparison to the state-of-the-art or any other work in the field. For a typical explainable AI work, there should be comparison with other existing explainability methods and demonstrate how the proposed work is superior. It is unclear how the performance of the proposed method really differ from the regular Grad-CAM in general.
Some of the baselines for comparison can be considered: LIME, Grad-CAM, GNN-Explainer, Attention-based methods etc.
3. In section 4.3 discussion of dataset III and IV. it is unclear how the patterns of brain activations are 'similar' when the topography plots are clearly different. Even if the topography plots are similar, the Chinese characters and English words have different meaning so it is not possible to justify there is common cognitive processing mechanisms between the two languages in this case.
4. In section 5.2, the channel selection results have high variations, the 20% increase for subject 6 is not generalizable to other subjects or datasets and there is no significance measurement for the effect of channel selection. It is unclear how effective or ineffective the channel selection method is.
5. There is a lack of ablation studies to prove the importance of those channels-identified. For instance, if those channels were removed, there should be a significant drop of classification performance.

**Questions:**

see weaknesses

---

### Official Review · Reviewer_chCL · 2024-11-09

**Soundness:** 2
**Presentation:** 3
**Contribution:** 2
**Rating:** 3
**Confidence:** 4

**Summary:**

This paper proposes the Grad-TopoCAM for enhancing the interpretability of deep learning-based EEG decoding models. It maps the gradients of feature maps to specific brain regions and facilitates channel selection across different EEG tasks. The proposed method is validated on eight DL models and four public datasets. Experimental results demonstrate its effectiveness.

**Strengths:**

1. The proposed model can be integrated into different EEG decoding models to enhance their interpretability. It is a universal interpretability and visualization method.
2. The proposed model has been validated on various DL methods and datasets.

**Weaknesses:**

1. Grad-CAM has been widely adopted for feature visualization including for EEG decoding models. The contributions of the proposed method compared with other visualization methods are not clear.
2. The proposed Grad-TopoCAM is employed for visualization analysis on the model with the highest accuracy for each subject. However, it’s noticed that the visualized features can be very different across subjects. In addition to the individual variability, are the learned features related to the models? Is it a fair comparison for the features learned by different models?
3. Although visualization is important for interpreting results, the proposed method does not enhance decoding performance or provide unique neuroscience insights. The authors may consider either improving its methodological novelty or deepening its neuroscience contributions.

**Questions:**

Please refer to the weaknesses.

---

### Meta-Review · Area_Chair_w2ut · 2024-12-13

**Metareview:**

There is a clear agreement between reviewers that the paper lacks novelty and does not provide convincing results.

Also the authors did not attempt to engage in the discussion.

For these reasons this work cannot be endorsed for publication at ICLR 2025.

**Additional Comments On Reviewer Discussion:**

This is a clear unanimous concerns among reviewers (problematic experimental results, high variance metrics, lack of ablation study, missing details on hyperparameters setting, quality of the writing)

---

### Decision · Program_Chairs · 2025-01-22

Reject